# One-Hour Post-Load Plasma Glucose and Altered Glucometabolic Profile in Youths with Overweight or Obesity

**DOI:** 10.3390/ijerph20115961

**Published:** 2023-05-26

**Authors:** Procolo Di Bonito, Giuliana Valerio, Maria Rosaria Licenziati, Domenico Corica, Malgorzata Wasniewska, Anna Di Sessa, Emanuele Miraglia del Giudice, Anita Morandi, Claudio Maffeis, Enza Mozzillo, Valeria Calcaterra, Francesca Franco, Giulio Maltoni, Maria Felicia Faienza

**Affiliations:** 1Department of Internal Medicine, “S. Maria delle Grazie” Hospital, 80078 Pozzuoli, Italy; 2Department of Movement Sciences and Wellbeing, University of Napoli “Parthenope”, 80133 Napoli, Italy; 3Neuro-Endocrine Diseases and Obesity Unit, Department of Neurosciences, Santobono-Pausilipon Children’s Hospital, 80139 Napoli, Italy; 4Department of Human Pathology in Adulthood and Childhood, University of Messina, 98125 Messina, Italy; 5Department of Woman, Child and of General and Specialized Surgery, University of Campania “Luigi Vanvitelli”, 80138 Napoli, Italy; 6Department of Surgery, Dentistry, Pediatrics and Gynecology, Section of Pediatric Diabetes and Metabolism, University and Azienda Ospedaliera Universitaria Integrata of Verona, 37126 Verona, Italy; 7Section of Pediatrics, Department of Translational Medical Science, Regional Center of Pediatric Diabetes, University of Naples “Federico II”, 80131 Napoli, Italy; 8Pediatric Department, “V. Buzzi” Children’s Hospital, 20154 Milano, Italy; 9Department of Internal Medicine, University of Pavia, 27100 Pavia, Italy; 10Pediatric Department, Azienda Sanitaria Universitaria Friuli Centrale, Hospital of Udine, 33100 Udine, Italy; 11Pediatric Unit, IRCCS Azienda Ospedaliero-Universitaria di Bologna, 40138 Bologna, Italy; 12Department of Precision and Regenerative Medicine and Ionian Area, University of Bari “Aldo Moro”, 70121 Bari, Italy

**Keywords:** one-hour post-load plasma glucose, obese, youth, glucometabolic, impaired glucose tolerance

## Abstract

In youths, two cut-offs (133 and 155 mg/dL) have been proposed to identify high glucose levels at the 1 h (G_60_) mark during an oral glucose tolerance test (OGTT). We evaluated which cut-off was more closely associated with isolated impaired glucose tolerance (IGT) and cardiometabolic risk (CMR) in 1199 youth with overweight/obesity (OW/OB) and normal fasting glucose and/or HbA1c. The disposition index (DI) was available in 724 youths. The sample was divided by two cut-offs of G_60_: <133 mg/dL (n = 853) and ≥133 mg/dL (n = 346), or G_60_ < 155 mg/dL (n = 1050) and ≥155 mg/dL (n = 149). Independent of the cut-off, youths with high levels of G_60_ showed higher levels of G_120_, insulin resistance (IR), triglycerides to HDL ratio (TG/HDL), alanine aminotransferase (ALT), and lower insulin sensitivity (IS) and DI than youths with lower levels of G_60_. The percentage of youths showing IGT, IR, low IS, high TG/HDL ratio, high ALT, and low DI was 50% higher in the G_60_ ≥ 133 mg/dL group than in the G_60_ ≥ 155 mg/dL one. In youths with OW/OB and IGT, a cut-off of G_60_ ≥ 133 mg/dL is more useful than G_60_ ≥ 155 mg/dL to identify those at high risk of IGT and altered CMR profile.

## 1. Introduction

In recent decades, with the rising prevalence of obesity (OB) in pediatric age, the incidence of associated cardiometabolic comorbidities has also increased [1]. Glucose dysregulation ranging from prediabetes to overt type 2 diabetes (T2DM) has often been observed in youth with overweight (OW) or OB [2].

Prediabetes is a condition characterized by a concentration of glucose above the normal level but lower than values that identify overt diabetes. It encompasses both impaired glucose tolerance and impaired fasting glucose [3]. A recent meta-analysis reported that obese subjects have a three times higher prevalence of prediabetes than non-diabetic individuals. Additionally, T2DM, prediabetes, and indices of glucose dysmetabolism are positively correlated with body mass index (BMI) in obese children and adolescents [4].

Evidence suggests that prediabetes influences the development of atherosclerotic cardiovascular diseases (CVDs) in adults [5]. On the other hand, young individuals affected by T2DM display an accelerated decline of insulin secretion and are at risk of developing related complications; hence, preemptive risk-based screening to detect asymptomatic subjects is crucial to quickly manage the condition [3].

The oral glucose tolerance test (OGTT) represents the gold standard for the diagnosis of glucose dysregulation so that individuals at high risk can be referred for lifestyle intervention to prevent the progression to T2DM and associated complications. 

The diagnosis of prediabetes, according to the American Diabetes Association (ADA), is based on the assessment of fasting glucose (G_0_), two-hour plasma glucose after OGTT (G_120_), or glycosylated hemoglobin (HbA1c) [3]. These methods differ in sensitivity and specificity, so they recognize distinct phenotypes with different risks of progression to overt T2DM [3]. 

Previous data suggest that in 30 to 40% of the T2DM-affected subjects, there is no evidence of any indicator suggesting the presence of the intermediate condition of impaired fasting glucose (IFG) and/or impaired glucose tolerance (IGT) [6].

Recently, the one-hour post-load plasma glucose (G_60_) ≥ 155 mg/dL (8.6 mmol/L) has been demonstrated to be highly prognostic for alterations of β-cell function and progression to overt T2DM in adults with normal glucose tolerance (NGT) [7,8]. Furthermore, this threshold seems more predictive of cardiovascular events than fasting or two-hour plasma glucose [9]. From previous studies, it appears that the G_60_ level may help identify individuals at risk of diabetes better than traditional glucose tests [10,11].

Experts from the European Association for the Study of Diabetes (EASD) [7] have raised a petition to associate G_60_ levels above 155 mg/dl with altered glucose tolerance in adults on the grounds that G_60_ is a more time-effective diagnostic marker for prediabetes risk.

The relevance of G_60_ for identifying individuals at high risk for IGT or abnormal cardiometabolic risk (CMR) profile has also been previously suggested in children with OB using the same cut-off proposed in adults [12,13,14,15]. However, the assessment of the optimal cut-off value of G_60_ in young people with OB might be a goal to identify, to an acceptable level of accuracy, individuals at risk of developing prediabetes/T2DM and CMR factors.

Indeed, Manco et al. found that the best G_60_ cut-off associated with IGT corresponded to a value ≥ 133 mg/dL (7.4 mmol/L) in youths with OB. They also demonstrated that this cut-off was closely associated with low insulin sensitivity and altered β-cell function [16]. Subsequently, in a prospective study conducted in a multi-ethnic cohort of youths with OB and NGT, Tricò et al. confirmed that this threshold is associated with a worse clinical and metabolic phenotype, characterized by alterations in insulin sensitivity (IS), β-cell function, and insulin clearance [17]. In addition, they demonstrated that G_60_ ≥ 133 mg/dL might be an independent predictor of progression to prediabetes. Regarding the association with CMR, Manco et al. [16] demonstrated higher levels of triglycerides in youths with G_60_ ≥ 133 mg/dL compared to the group < 133 mg/dL, while this finding was not confirmed by Marcovecchio et al. [18]. 

The aim of this cross-sectional multicenter study was to compare the performance of G_60_ cut-off ≥133 mg/dL and ≥155 mg/dL to identify IGT, impaired insulin and β-cell function, and altered CMR profile in young people with OW/OB and normal levels of fasting glucose and HbA1c. 

## 2. Materials and Methods

### 2.1. Study Design, Study Site, and Study Population

This is a cross-sectional study conducted on behalf of the “Childhood Obesity” study group of the Italian Society for Pediatric Endocrinology and Diabetology (ISPED). Nine Italian centers provided data from the clinical records of 1199 non-diabetic children and adolescents (mean age 11.7 years) with OW/OB consecutively observed in the period of June 2016–June 2020, as previously described [19].

For the purpose of this present study, youths with impaired fasting glucose or HbA1c ≥ 5.7%, defined according to the ADA criteria [3], were excluded. Other exclusion criteria were systemic and endocrine diseases and use of medications affecting glucose metabolism. 

This study was conducted according to the principles expressed in the Declaration of Helsinki. Written informed consent was obtained from the parents or tutors of all participants. This study was approved by the Ethics Committee of the AORN Santobono-Pausilipon (reference number 22877/2020).

### 2.2. Anthropometric, Clinical, and Biochemical Variables

Height and weight were measured for each patient using standard techniques. BMI was calculated as the ratio weight (kg)/height^2^ (m^2^) and transformed to age- and sex-specific standard deviation score (SDS) on the basis of reference data from Italian BMI percentiles [20].

Blood pressure (BP) was measured three times 2 min apart, in a sitting position after 5 min of rest, using aneroid sphygmomanometers with appropriately sized cuffs, according to standard procedures. The mean of the last two values was used [21].

Biochemical analyses were performed in the centralized laboratory of each center. All laboratories belong to the Italian National Health System and are certified according to International Standards ISO 9000 (www.iso9000.it/ accessed on 14 November 2022), undergoing semi-annual quality controls and inter-lab comparisons.

After overnight fasting, biochemical markers of lipid metabolism and alanine-aminotransferase (ALT) were analyzed. Triglycerides (TG) to HDL-cholesterol ratio was also calculated (TG/HDL ratio). HbA1c was assessed via high-performance liquid chromatography as elsewhere described [2]. OGTT was performed using 1.75 g/kg of glucose up to a maximum of 75 g [3,22]. Glucose and insulin levels at baseline (G_0_, I_0_), one-hour (G_60_), and two-hour post-load glucose (G_120_) were analyzed. Data on glucose and insulin at 30′ during OGTT were available in a subsample of 724 youths, as previously described [2]. 

Insulin resistance (IR) was measured using the homeostasis model assessment (HOMA-IR) index, calculated as insulin (μU/mL) × fasting glucose (mmol/L)/22.5 [23].

The insulin sensitivity (IS) was calculated as 1/ insulin at time 0 (1/I_0_) [24]. Insulinogenic index (IGI) was calculated as Δ(I_0_ − I_30_)/Δ(G_0_ − G_30_), where insulin was expressed as µU/mL and glucose as mg/dL. DI was calculated via the following formula: IGI × 1/I_0_, as described elsewhere [25]. 

### 2.3. Definition of Variables

OW and OB were defined on the basis of the finding of BMI ≥ 75th and ≥95th percentile, respectively, in accordance with the Italian BMI standards [18]. IR was defined using the 97th percentile of HOMA-IR distribution by age and gender in normal-weight Italian children [26]. Low IS or low DI was defined by the 25th percentile of, respectively, 1/I_0_ or DI as calculated in our sample [2]. High TG/HDL ratio was defined by the 75th percentile of TG/HDL ratio distribution in our sample. High ALT levels, as surrogate of nonalcoholic fatty liver disease (NAFLD), were defined using a cut-point > 25.8 IU/L in boys and 22.1 IU/L in girls, corresponding to the 95^th^ percentile level for ALT in a healthy, normal weight pediatric population [27].

### 2.4. Statistical Analysis

Continuous data were expressed as mean ± standard deviation (SD), numbers and proportions as percentages (%), and 95% confidence interval (CI). Variables with skewed distribution (i.e., HOMA-IR, 1/I_0_, ALT, and DI) were log-transformed for the analysis and expressed as median and interquartile range. Mean values were compared using Student’s *t* test. Distribution of categories was compared by χ^2^, and, when needed, exact tests were performed using the Monte Carlo method.

The relationships between two cut-offs of G_60_ (133 mg/dL or 155 mg/dL) with IGT, IR, low IS, high TG/HDL ratio, and ALT were analyzed using receiver operator curve (ROC) analysis. The area under curve (AUC) was obtained using IGT or the other metabolic variables as dependent variables and G_60_ ≥ 133 mg/dL or G_60_ ≥ 155 mg/dL as variables of interest. Sensitivity and specificity were calculated using 2 × 2 tables.

A *p* value < 0.05 was considered statistically significant. The statistical analysis was performed using IBM SPSS Statistics, Version 20.0. Armonk, NY, USA.

## 3. Results

A total of 1199 young people aged 5–18 years were included in this study. Of these, 595 were boys, and 604 were girls. The characteristic of the study population by sex is reported in Table 1. Boys showed slightly higher levels of fasting G_0_, ALT, and systolic BP and lower levels of IS compared to girls. The prevalence of isolated IGT did not significantly differ between boys and girls (5.4% vs. 7.6%, *p* = 0.116). 

In the overall sample, the prevalence of youths with G_60_ ≥ 133 mg/dL was 28.9% vs. 12.4% in youths with G_60_ ≥ 155 mg/dL (*p* < 0.0001). The features of the study sample divided by the two G_60_ cut-offs are reported in Table 2. 

Youths with G_60_ ≥ 133 mg/dL showed higher levels of G_0_, G_60_, G_120_, HbA1c, HOMA-IR, TG, HDL-C, TG/HDL-C, ALT, and lower IS and DI than the group with G_60_ < 133 mg/dL. Youths with G_60_ ≥ 155 mg/dL showed a similar profile to those with G_60_ < 155 mg/dL, except for the G_0_.

After removing youths with G_60_ ≥ 155 mg/dL, 1050 young people were reclassified according to the cut-off of G_60_ ≥ 133 mg/dL. Their features are reported in Table 3. 

In this subsample, individuals with G_60_ ≥133 mg/dL showed similar features to that observed in the whole sample in terms of parameters of insulin, β-cell function, and CMR factors. 

In the overall sample, the AUC (95%Cl) of G_60_ as a continuous variable was 0.86 (0.82–0.89) (*p* < 0.0001) compared to the isolated IGT. The G_60_ ≥ 133 mg/dL value showed an AUC of 0.76 (0.71–0.82) (*p* < 0.0001), while the cut-off at G_60_ ≥ 155 mg/dL showed an AUC of 0.75 (0.68–0.82) (*p* < 0.0001).

The percentage of youths with IGT, IR, low IS, high TG/HDL ratio, high ALT, and low DI according to the cut-offs of G_60_ ≥ 133 mg/dL or ≥155 mg/dL was significantly higher than their respective counterparts of G_60_ < 133 mg/dL or <155 mg/dL (Figure 1).

The performance of the two cut-offs at G_60_ in relation to IGT, metabolic abnormalities, and CMRF is reported in Table 4. For each factor analyzed, the cut-off at G_60_ ≥ 133 mg/dL showed higher sensitivity and lower specificity with respect to G_60_ ≥ 155 mg/dL. 

## 4. Discussion

This study demonstrates that in young people with OW/OB and normal fasting glucose and HbA1c, a cut-off of G_60_ ≥ 133 mg/dL is more useful than G_60_ ≥ 155 mg/dL to identify individuals with isolated IGT, β-cell impairment, and altered CMR profile.

Prediabetes in youth has become a more frequent challenge for patients, families, and healthcare professionals. Knowledge of the natural history of prediabetes and T2DM have demonstrated differences between these conditions in juveniles and adults, making extrapolation from adult practice problematic. Furthermore, guidelines for the management and treatment of prediabetes in young people are lacking. The screening of prediabetes or T2DM in childhood is controversial. A recent statement from “The US Preventive Services Task Force” concludes that the current evidence is insufficient to assess the balance of benefits and harms of screening for T2DM in children and adolescents [28]. In contrast, the ADA recommends screening for prediabetes/diabetes in youths with at least one risk factor among obesity, family history of diabetes, gestational diabetes, and other conditions associated with IR [3]. 

The one-hour plasma glucose during OGTT seems to be a useful early biomarker of glucose dysregulation [29]. A cut-off value of ≥155 mg/dl (8.6 mmol/L) was initially identified in the San Antonio Heart Study [30]. This finding was subsequently confirmed in the Botnia Study that demonstrated a better predictive power of the one-hour post-load plasma glucose for the risk of developing T2DM than fasting and post-load glucose at 120’ fasting glucose [31]. Several longitudinal studies have confirmed that individuals with NGT and one-hour post-load plasma glucose value ≥ 155 mg/dl (≥8.6 mmol/L) are at increased risk for T2DM [32,33,34,35,36]. 

Although a value of G_60_ ≥ 155 has been shown to be useful in predicting progression to prediabetes and β-cell deterioration in Latino youths with OB and a family history of T2DM [13], subsequent studies assessed the performance of diverse values of G_60_. For instance, using ROC analysis, Manco et al. demonstrated that the cut-off of G_60_ ≥ 133 mg/dL was able to identify IGT with higher sensitivity and specificity than G_60_ ≥ 155 mg/dL in Caucasian youths with OB [16]. The efficacy of this lower cut-point in predicting the progression to prediabetes was subsequently confirmed in a prospective study in normoglycemic youths with OB [17]. 

Recently, an international panel of experts supported a petition for redefining the current diagnostic criteria for prediabetes with the use of the elevated one-hour post-load plasma glucose level [7]. Given the paucity of pediatric studies on this issue, it is useful to provide more evidence about the most accurate cut-off of G_60_ for the screening of prediabetes among youths. Compared to the study by Manco et al., we obtained a similar performance in terms of accuracy to identify IGT using the G_60_ cut-off ≥ 133 mg/dL (sensitivity 0.81 vs. 0.78 and specificity 0.74 vs. 0.75, respectively) or the G_60_ ≥ 155 mg/dL (sensitivity 0.45 vs. 0.59 and specificity 0.91 vs. 0.91, respectively).

In our sample, 88.4% of youths showed at least one risk factor for diabetes, i.e., elevated BP, high TG, high ALT, low HDL-cholesterol, or a family history of T2DM. Indeed, according to the consensus position statement of the ISPED (that implemented the ADA guidelines), the presence of several associated cardiovascular risk factors suggests the assessment of OGTT in youths with OB [22].

We extended our comparative analysis regarding the accuracy of the two thresholds to identify altered insulin and β-cell function and abnormal CMR profile. The main finding of our study supported the higher predictive ability of G_60_ ≥ 133 mg/dL to detect individuals with several metabolic abnormalities and a worse CMR profile. This latter association was demonstrated only for higher levels of triglycerides by Manco et al. [16] but not by Marcovecchio et al. [18]. This discrepancy may be explained by the exclusion of individuals with IFG and IGT by the latter. This finding extends previous studies that confirmed the association between prediabetes, particularly IGT [2] and CMR [37,38]. In our study, the high predictive ability of G_60_ ≥ 133 mg/dL was confirmed also in the subsample of youths who were reclassified after the removal of individuals with G_60_ ≥ 155 mg/dL. The strong association between G_60_ ≥ 133 mg/dL and CMR factors contributes to strengthening the greater usefulness of this cut-off value in children and adolescents compared to that derived from adults. 

Although the determinations of fasting glucose and HbA1c are the most frequently used tests for screening of prediabetes/diabetes in the pediatric population [39], the assessment of G_60_ during the OGTT in children or adolescents might be considered an early biomarker of glucose metabolism impairment compared to the complete OGTT. Another consideration in favor of one-hour post-load glucose is relative to the difficulties in fulfilling the guidelines about sample centrifugation, storage, and transport in ice [40]. Delays in handling and analysis of samples might increase the risk of underestimation of prediabetes/diabetes [41]. Interestingly, the one-hour post-load plasma glucose level, among the high-risk OGTT-glucose phenotypes, represents a good biomarker in response to lifestyle intervention [42].

Concerning the cost-effectiveness of performing 1 h afterload glucose in obese subjects as a screening strategy to identify prediabetes, the data are heterogeneous as the success of a screening program depends on the context (epidemiology, social factors, political priorities, and budget constraints). A recent literature review shows that more cost-effective evaluations of national and regional prevention programs for non-communicable diseases are needed to guide policymakers [43]. We believe that good clinical practice would be to implement screening for young people at risk for prediabetes and lifestyle modification through improved nutrition and exercise. In some cases, pharmacological intervention may even be warranted, but always in the context of lifestyle and behavioral changes.

### Limitation and Strength 

Limitations of this study include the following: (1) the cross-sectional observational nature that precludes evaluation of the progression over time of children with the G_60_ ≥ 133 mg/dL; (2) the evaluation of a sample of youths with OW/OB that limits the extendibility of our results to the general population; (3) the derivation of the measures to estimate insulin sensitivity and β-cell function from the OGTT.

The strength of our study includes a large sample of young people without impaired fasting glucose and HbA1c ≥ 5.7% and the evaluation of metabolic and CMR profiles.

## 5. Conclusions

Our results confirm and extend the findings of previous studies about the use of one-hour post-load glucose in identifying children and adolescents with isolated IGT and metabolic abnormalities. An elevated one-hour post-load plasma glucose level, not measured with current diagnostic standards, may provide an opportunity for the early identification of a large population despite normal levels of fasting glucose and HbA1c. The finding of an elevated one-hour post-load plasma glucose level can allow lifestyle intervention that has the greatest benefit for preserving or reversing β-cell function and preventing further progression to prediabetes and diabetes.

We suggest that the higher accuracy in terms of sensitivity of a lower cut-off compared to that proposed in adults is more appropriate to identify youths at risk of altered glucose metabolism. Further studies are needed to assess the predictive value of the cut-off ≥133 mg/dL on the monitoring of the cardiometabolic health of youths with OB. 

## Figures and Tables

**Figure 1 ijerph-20-05961-f001:**
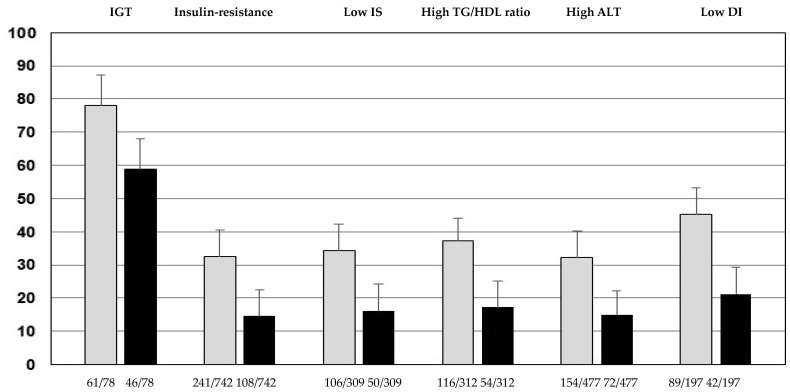
Percentage of youths with impaired glucose tolerance (IGT), insulin resistance, low insulin sensitivity (IS), high triglycerides to HDL-cholesterol ratio (TG/HDL ratio), high alanine aminotransferase (ALT), and low disposition index (DI) in youths with G_60_ ≥ 133 mg/dL (grey bars) or G_60_ ≥ 155 mg/dL (black bars) (*p* < 0.0001 compared to their respective counterparts < 133 mg/dL or G_60_ < 155 mg/dL). Youths in the G_60_ ≥ 133 mg/dL group showed an increase of 33% for IGT, 123% for IR, 104% for low IS, 112% for high TG/HDL ratio, 114% for high ALT, and 103% for low DI, with respect to individuals in the G_60_ ≥ 155 mg/dL group.

**Table 1 ijerph-20-05961-t001:** Characteristics of the overall sample and by sex.

	Overall Sample	Boys	Girls	*p* Value
*n*	*1199*	*595*	*604*	
Age (years)	11.6 ± 2.6	11.6 ± 2.5	11.6 ± 2.8	0.745
BMI (kg/m^2^)	30.7 ± 5.4	30.8 ± 5.3	30.5 ± 5.5	0.330
BMI-SDS	2.3 ± 0.6	2.3 ± 0.6	2.3 ± 0.6	0.706
G_0_, (mg/dL)	86.0 ± 8.1	86.8 ± 7.9	85.3 ± 8.2	0.001
G_60_, (mg/dL)	122.6 ± 28.0	124.0 ± 28.9	121.2 ± 27.1	0.095
G_120_, (mg/dL)	108.4 ± 19.5	108.9 ± 18.2	107.9 ± 20.7	0.367
HOMA-IR	3.8 (2.5–5.5)	3.4 (2.4–4.8)	3.7 (2.4–5.2)	0.169
1/I_0_ (μU/mL)	0.06 (0.04–0.08)	0.06 (0.05–0.09)	0.06 (0.04–0.09)	0.050
TC (mg/dL)	154.2 ± 29.4	153.7 ± 29.1	154.6 ± 28.9	0.597
HDL-C (mg/dL)	47.8 ± 10.4	47.7 ± 10.3	47.9 ± 10.6	0.711
Triglycerides (mg/dL)	81.0 (63.0–107.0)	77.0 (60.0–102.0)	79.0 (62.0–104.0)	0.330
TG/HDL ratio	1.8 (1.3–2.5)	1.7 (1.2–2.4)	1.7 (1.3–2.3)	0.516
SBP (mmHg)	113.3 ± 13.9	114.4 ± 13.4	112.2 ± 14.2	0.005
DBP (mmHg)	67.5 ± 9.4	67.6 ± 9.3	67.4 ± 9.4	0.652
ALT (IU/mL)	22.0 (16.0–31.0)	23.0 (17.0–34.0)	20.0 (15.0–27.0)	<0.0001
DI (n = 724)	0.15 (0.10–0.23)	0.15 (0.10–0.23)	0.15 (0.10–0.23)	0.702
IGT (%)	78 (6.5)	32 (5.4)	46 (7.6)	0.116

Data are expressed as mean ± standard deviation or number (%) or median and interquartile range. ALT: alanine aminotransferase; BMI: body mass index; DBP: diastolic blood pressure; DI: disposition index; G_0_: fasting glucose; G_60_: 1 h glucose; G_120_: 2 h glucose; HDL-C: high-density lipoprotein cholesterol; HOMA-IR: homeostasis model assessment; IGT: impaired glucose tolerance; 1/I_0_: 1/insulin at time 0; SBP: systolic blood pressure; TC: total cholesterol; TG/HDL ratio: triglycerides to HDL-cholesterol ratio.

**Table 2 ijerph-20-05961-t002:** Features of children and adolescents without IFG and high HbA1c according to different cut-offs of G_60_.

	G_60_ < 133 mg/dL	G_60_ ≥ 133 mg/dL	*p* Value	G_60_ < 155 mg/dL	G_60_ ≥ 155 mg/dL	*p* Value
*n = 1199 (100%)*	*853 (71.1)*	*346 (28.9)*		*1050 (87.6)*	*149 (12.4)*	
Age (years)	11.5 ± 2.7	11.8 ± 2.5	0.054	11.5 ± 2.7	11.9 ± 2.4	0.160
Girls, n (%)	438 (51)	166 (48)	0.290	538 (51)	66 (44)	0.113
BMI (kg/m^2^)	30.6 ± 5.2	30.9 ± 5.9	0.305	30.7 ± 5.4	30.4 ± 5.4	0.561
BMI-SDS	2.3 ± 0.6	2.3 ± 0.6	0.826	2.3 ± 0.6	2.3 ± 0.6	0.174
G_0_, (mg/dL)	85.7 ± 7.7	87.0 ± 8.8	0.008	86.0 ± 7.9	86.6 ± 9.3	0.346
G_60_, (mg/dL)	107.5 ± 16.3	155.2 ± 18.5	<0.0001	114.8 ± 20.6	171.8 ± 16.0	<0.0001
G_120_, (mg/dL)	103.5 ± 15.7	120.3 ± 22.4	<0.0001	106.3 ± 17.0	123.0 ± 27.8	<0.0001
HOMA-IR	3.4 (2.3–4.9)	4.0 (2.7–5.6)	<0.0001	3.4 (2.3–4.9)	4.1 (2.8–6.1)	<0.0001
1/I_0_ (μU/mL)	0.06 (0.04–0.09)	0.05 (0.04–0.08)	<0.0001	0.06 (0.04–0.09)	0.05 (0.04–0.07)	<0.0001
TC (mg/dL)	153.4 ± 29.9	156.2 ± 28.3	0.134	153.7 ± 29.5	157.5 ± 28.8	0.147
HDL-C (mg/dL)	48.1 ± 10.5	46.9 ± 10.2	0.077	47.9 ± 10.5	46.9 ± 9.8	0.267
TG (mg/dL)	76.0 (60.0–100.0)	87.0 (63.0–112.0)	0.002	77.0 (61.0–102.0)	88.0 (65.0–112.0)	0.033
TG/HDL ratio	1.6 (1.2–2.2)	1.9 (1.3–2.6)	0.001	1.7 (1.2–2.3)	1.9 (1.4–2.5)	0.031
SBP (mmHg)	113.3 ± 14.0	113.2 ± 13.6	0.932	113.4 ± 13.9	112.7 ± 13.8	0.571
DBP (mmHg)	67.7 ± 9.6	67.0 ± 8.7	0.246	67.5 ± 9.4	67.2 ± 9.0	0.717
ALT (IU/mL)	21.0 (15.0–28.0)	22.0 (16.0–33.0)	0.002	21.0 (16.0–29.0)	24.0 (17.0–34.0)	0.008
DI (n = 724)	0.17 (0.11–0.26)	0.11 (0.08–0.15)	<0.0001	0.16 (0.10–0.24)	0.10 (0.07–0.14)	<0.0001

Data are expressed as mean± standard deviation or number (%) or median and interquartile range ALT: alanine aminotransferase; BMI: body mass index; DBP: diastolic blood pressure; DI: disposition index; G_0_: fasting glucose; G_60_: 1 h glucose; G_120_: 2 h glucose; HDL-C: high-density lipoprotein cholesterol; HOMA-IR: homeostasis model assessment; 1/I_0_: fasting insulin; SBP: systolic blood pressure; TC: total cholesterol; TG/HDL ratio: triglycerides to HDL-cholesterol ratio.

**Table 3 ijerph-20-05961-t003:** Characteristics of youths with G_60_ <155 mg/dL reclassified by cut-off of 133 mg/dL.

	G_60_ < 133 mg/dL	G_60_ ≥ 133 mg/dL	*p* Value
*n = 1050*	*853*	*197*	
Age (years)	11.5 ± 2.7	11.8 ± 2.6	0.178
Girls, n (%)	438 (51)	100 (51)	0.882
BMI (kg/m^2^)	30.6 ± 5.2	31.3 ± 6.2	0.089
BMI-SDS	2.3 ± 0.6	2.4 ± 0.6	0.201
G_0_, (mg/dL)	85.7 ± 7.7	87.3 ± 8.4	0.008
G_60_, (mg/dL)	107.5 ± 16.3	142.7 ± 6.2	<0.0001
G_120_, (mg/dL)	103.5 ± 15.7	118.3 ± 17.1	<0.0001
HOMA-IR	3.4 (2.3–4.9)	3.8 (2.7–5.3)	0.005
1/I_0_ (μU/mL)	0.06 (0.04–0.09)	0.05 (0.04–0.08)	0.013
TC (mg/dL)	153.4 ± 30.0	155.2 ± 27.9	0.428
HDL-C (mg/dL)	48.1 ± 10.5	46.7 ± 10.5	0.171
Triglycerides (mg/dL)	76.0 (60.0–100.0)	85.0 (63.0–112.0)	0.019
TG/HDL ratio	1.6 (1.2–2.2)	1.9 (1.3–2.5)	0.011
SBP (mmHg)	113.3 ± 14.0	113.6 ± 13.4	0.757
DBP (mmHg)	67.7 ± 9.6	66.8 ± 8.4	0.243
ALT (IU/mL)	21.0 (15.0–28.0)	21.0 (16.0–31.5)	0.059
DI (n = 642)	0.17 (0.11–0.26)	0.12 (0.09–0.16)	<0.0001

Data are expressed as mean ± standard deviation or number (%) or median and interquartile range ALT: alanine-aminotransferase; BMI: body mass index; DBP: diastolic blood pressure; DI: disposition index; G_0_: fasting glucose; G_60_: 1 h glucose; G_120_: 2 h glucose; HDL-C: high-density lipoprotein cholesterol; HOMA-IR: homeostasis model assessment; IGT: impaired glucose tolerance; 1/I_0_: 1/insulin at time 0; SBP: systolic blood pressure; TC: total cholesterol; TG/HDL ratio: triglycerides to HDL-cholesterol ratio.

**Table 4 ijerph-20-05961-t004:** Performance of G_60_ ≥ 133 mg/dL and ≥155 mg/dL in relation to IGT, insulin resistance, low insulin sensitivity, high TG/HDL ratio, high ALT, and low disposition index.

Factors	Sensitivity	Specificity	PPV	NPV
IGT				
G_60_ ≥ 133 mg/dL	0.78 (0.76–0.81)	0.75 (0.72–0.77)	0.18 (0.16–0.20)	0.98 (0.97–0.99)
G_60_ ≥ 155 mg/dL	0.59 (0.56–0.62)	0.91 (0.89–0.92)	0.31 (0.28–0.34)	0.97 (0.96–0.98)
Insulin-resistance				
G_60_ ≥ 133 mg/dL	0.33 (0.30–0.35)	0.77 (0.75–0.79)	0.70 (0.67–0.72)	0.41 (0.39–0.44)
G_60_ ≥ 155 mg/dL	0.15 (0.13–0.17)	0.91 (0.89–0.93)	0.73 (0.70–0.75)	0.40 (0.37–0.43)
Low insulin-sensitivity				
G_60_ ≥ 133 mg/dL	0.34 (0.32–0.37)	0.73 (0.70–0.76)	0.31 (0.28–0.33)	0.76 (0.74–0.79)
G_60_ ≥ 155 mg/dL	0.16 (0.14–0.18)	0.89 (0.87–0.91)	0.34 (0.31–0.36)	0.75 (0.73–0.78)
High TG/HDL ratio				
G_60_ ≥ 133 mg/dL	0.37 (0.34–0.40)	0.74 (0.72–0.77)	0.34 (0.31–0.36)	0.77 (0.75–0.79)
G_60_ ≥ 155 mg/dL	0.17 (0.15–0.20)	0.89 (0.87–0.91)	0.36 (0.34–0.39)	0.75 (0.73–0.78)
High ALT				
G_60_ ≥ 133 mg/dL	0.32 (0.30–0.35)	0.73 (0.71–0.76)	0.45 (0.42–0.47)	0.62 (0.59–0.65)
G_60_ ≥ 155 mg/dL	0.15 (0.13–0.17)	0.89 (0.87–0.91)	0.48 (0.46.0.51)	0.61 (0.59–0.64)
Low disposition index				
G_60_ ≥ 133 mg/dL	0.45 (0.42–0.49)	0.74 (0.71–0.78)	0.40 (0.36–0.43)	0.78 (0.75–0.81)
G_60_ ≥ 155 mg/dL	0.21 (0.18–0.25)	0.92 (0.90–0.94)	0.51 (0.48–0.55)	0.76 (0.73–0.79)

ALT: alanine aminotransferase; G_60_: 1 h glucose; IGT: impaired glucose tolerance; NPV: negative predictive value; PPV: positive predictive value; TG/HDL ratio: triglycerides to HDL-cholesterol ratio_._

## Data Availability

Not applicable.

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
