# Peer review of "One-Hour Post-Load Plasma Glucose and Altered Glucometabolic Profile in Youths with Overweight or Obesity"

_ijerph, 2023, doi:10.3390/ijerph20115961_

Round 1
Reviewer 1 Report
Compliments for your effort. The article represents a fundamental tool to select in overweigh and obese youth those at risk for adverse outcomes.
Please correct at line 38 "iGT"
and explain better at line 116 the ratio 1/Io, as 1/ insulin at time 0 as microU/ml
Author Response
Reviewer 1
Compliments for your effort. The article represents a fundamental tool to select in overweight and obese youth those at risk for adverse outcomes.
Please correct at line 38 "iGT"
Answer: we corrected the mistake
and explain better at line 116 the ratio 1/Io, as 1/ insulin at time 0 as microU/ml
Answer: We better specified this abbreviation both in the text and in the legends to tables.

Reviewer 2 Report
The paper entitled “One-hour post-load plasma glucose and altered glucometabolic profile in youths with overweight or obesity” is well structured and presented. The main topics of the paper are well introduced, and the methodology and results are well organized. This work showed the impact of a lower cut-off (133 mg/dl) on G60 of OGTT.
The authors must improve the legend of table 2, where IGT meaning is not necessary.
About table 4, I consider it a figure, and suggest to change to figure 1. The interpretation of this table results are not correct in respect to IGT, where the difference are not higher than 50% as referred in line 164 (about 60% in comparison to less than 80%).
Author Response
Reviewer 2
The paper entitled “One-hour post-load plasma glucose and altered glucometabolic profile in youths with overweight or obesity” is well structured and presented. The main topics of the paper are well introduced, and the methodology and results are well organized. This work showed the impact of a lower cut-off (133 mg/dl) on G60 of OGTT.
The authors must improve the legend of table 2, where IGT meaning is not necessary.
Answer: we thank the reviewer for this suggestion, that prompted us to revise also the legends of Tables 1, 3 and 4, where abbreviations were not listed in alphabetical order or were not complete.
About table 4, I consider it a figure, and suggest to change to figure 1. The interpretation of this table results are not correct in respect to IGT, where the difference are not higher than 50% as referred in line 164 (about 60% in comparison to less than 80%).
Answer: the reviewer is right, Table 4 was a “typo”, that has been amended along with the % differences between groups. The following sentences were added: The percentage of youths with IGT, IR, low IS, high TG/HDL ratio, high ALT and low DI according to the cut-offs of G60 ≥133 mg/dL or ≥155 mg/dL was significantly higher than their respective counterparts of G60 <133 mg/dL or <155 mg/dL (Figure). Youths in the G60 ≥133 mg/dL group showed an increase of 33% for IGT, 123% for IR, 104% for low IS, 112% for high TG/HDL ratio, 114% for high ALT and 103% for low DI, respect to individuals in the G60 ≥155 mg/dL group.

Reviewer 3 Report
The aim of this cross-sectional multicenter study is to compare the performance of G60 cut-off ≥133 mg/dL and ≥155 mg/dL to identify IGT, impaired insulin and β-cell function, and altered CMR profile in young people with OW/OB and normal levels of fasting glucose and HbA1c.
They have already published similar data in another paper. https://doi.org/ 10.3390/ijerph20020928
The study was conducted formally well.
There are some doubts to be clarified.
1. What does this paper add to the already published data? Could not one paper have been published?
2. The rationale of the paper is unclear. What is the point of defining a new diagnostic cut-off in these subjects? What advantages would enlarging the number of subjects diagnosed with pre-diabetes offer? The therapy in an overweight or obese subject is allways the same. Should we be more incisive with drug therapy whose use is controversial in all patients with pre-diabetes?
3. One essential parameter was not evaluated. The abdominal circumference. This measurement together with familiarity are sufficient values to assess risk in these patients. To make a blood glucose curve for all overweight subjects is a huge cost, the benefit of which is unclear (see point 2).
4. The authors should clarify in the discussion what is the advantage of performing ONE-HOUR POST-LOAD PLASMA GLUCOSE a in these subjects and compare the cost-effectiveness of this diagnostic protocol with particular reference to social and educational policies for controlling childhood obesity.
5. The literature suggests that in addition to the blood glucose value after one hour, it is essential to assess the morphology of the curve. Morphologies of the glucose curve seem reflecting different metabolic phenotypes of insulin action and secretion, particularly when combined with morphologies of insulin curve or time of glucose peak. https://eje.bioscientifica.com/view/journals/eje/166/1/107.xml
6. Beware of plagiarised parts in the paper and whether all 14 authors actually participated in the work
14 autori?

Author Response
Reviewer 3
The aim of this cross-sectional multicenter study is to compare the performance of G60 cut-off ≥133 mg/dL and ≥155 mg/dL to identify IGT, impaired insulin and β-cell function, and altered CMR profile in young people with OW/OB and normal levels of fasting glucose and HbA1c.
They have already published similar data in another paper. https://doi.org/ 10.3390/ijerph20020928
The study was conducted formally well.
There are some doubts to be clarified.
- What does this paper add to the already published data? Could not one paper have been published?
Answer: we really thank to the reviewer for his/her observations. Our group has been working for a long time on the aspects of comorbidities related to overweight/obesity in children with particular regard to impaired glucose metabolism. The mentioned paper produced by our group (https://doi.org/10.3390/ijerph20020928) was aimed at assessing the most appropriate cut-off of HbA1c for prediabetes screening in Caucasian youths with overweight or obesity, starting from the assumption that this marker has been now included in the diagnostic work-up also in children and adolescents. The difference between this paper and the previous one relies upon the fat that in the current study participants had normal fasting glucose and HbA1c values.
Considering that this is a special issue (Diabetes: Screening, Prevention, Diagnosis and Therapy), we sought to explore this parallel topic, which has still open to debate between scholars, i.e. the predictive value of G60 respect to isolated IGT and altered cardiometabolic profile in these subjects. Indeed, the role of G60 to identify individuals with low insulin sensitivity, impaired beta-cell function, worse cardiovascular risk and higher mortality has been demonstrated in adults (see the following references):
- Bianchi C, et al. Elevated 1-hour post load plasma glucose levels identify subjects with normal glucose tolerance but impaired β-cell function, insulin resistance, and worse cardiovascular risk profile: the GENFIEV study. J Clin Endocrinol Metab. 2013 May;98(5):2100-5. doi: 10.1210/jc.2012-3971;
- Rong L, et al. One-hour plasma glucose as a long-term predictor of cardiovascular events and all-cause mortality in a Chinese older male population without diabetes: A 20-year retrospective and prospective study. Front Cardiovasc Med. 2022 Aug 22;9:947292. doi: 10.3389/fcvm.2022.947292;
- Liu X, et al. Reversion from Pre-Diabetes Mellitus to Normoglycemia and Risk of Cardiovascular Disease and All-Cause Mortality in a Chinese Population: A Prospective Cohort Study. J Am Heart Assoc. 2021 Feb 2;10(3):e019045. doi: 10.1161/JAHA.120.019045).
In particular, G60≥155 mg/dL among adult subjects with normal glucose tolerance, has been demonstrated to be predictive for developing T2DM and cardiovascular disease (Bianchi C, et al. 2013). Our paper demonstrates that, whenever the G60 would be accepted in the diagnostic work up in pediatric subjects with overweight/obesity, the cut-off of G60≥133 seems to be more sensitive than the cut-off derived from adults because it identifies a greater number of subjects at risk of IGT and adverse cardiovascular profile.
- The rationale of the paper is unclear. What is the point of defining a new diagnostic cut-off in these subjects? What advantages would be enlarging the number of subjects diagnosed with pre-diabetes offer? The therapy in an overweight or obese subject is always the same. Should we be more incisive with drug therapy whose use is controversial in all patients with pre-diabetes?
Answer: we thank to the reviews to allow us to clarify these aspects.
-Our study does not intend to introduce a "new cut-off" for G60, but to validate the cut-off proposed by Manco and Caprio in youths with overweight or obesity and normal fasting glycemia (References number 11 and 12 in the manuscript). Furthermore, Marcovecchio et al. failed to demonstrate that this cut-off identifies normoglycemic subjects with altered cardiometabolic risk (Reference number 13 in the manuscript). On the contrary, our study, using a large series of youths with overweight/obesity, was able to demonstrate that G60≥133 mg/dl identifies youths with a higher risk of IGT and adverse cardiometabolic profile. Furthermore, studies in adults have shown that G60 is more effective than 2-hour post load PG in predicting type 2 diabetes and its complications (Pareek M, et al. Enhanced Predictive Capability of a 1-Hour Oral Glucose Tolerance Test: A Prospective Population-Based Cohort Study. Diabetes Care. 2018 Jan;41(1):171-177. doi: 10.2337/dc17-1351).
-About the advantages of diagnosing a prediabetic condition in obese or overweight youths it should be considered that prediabetes represents a state of increased health risk that is defined by elevated blood glucose in addition to other health risks, such as high blood pressure, dyslipidemia, and other obesity-related conditions. Identifying patients with prediabetes (especially in post-load state, as surrogate of post-prandial state) has important benefits for individuals as well as for the health system. Indeed, an improved BMI trajectory after prediabetes identification was documented in youths with OW or OB followed longitudinally in a large academic-affiliated primary care network (Vajravelu ME, Lee JM, Shah R, Shults J, Amaral S, & Kelly A. Association between prediabetes diagnosis and body mass index trajectory of overweight and obese adolescents. Pediatr Diabetes. 2020; 21:743-746). Therefore, prediabetes screening may be beneficial beyond its intended goal of identification of glucose dysmetabolism.
-Prediabetes in youth has become a more frequent challenge for patients, families and healthcare professionals. The knowledge of the natural history of prediabetes and T2DM has demonstrated differences between these conditions in youth and adults, making extrapolation from adult practice problematic. Furthermore, guidelines for the management and treatment of prediabetes in young people are lacking. We think that a good clinical practice would be to individuate a simple screen method for young people at risk for prediabetes, as G60, and intervene with intensive lifestyle modification through improved nutrition and exercise. In some cases, pharmacological intervention may also be warranted, but always in the context of lifestyle and behavioral changes (We added these considerations in the discussion section
- One essential parameter was not evaluated. The abdominal circumference. This measurement together with familiarity are sufficient values to assess risk in these patients. To make a blood glucose curve for all overweight subjects is a huge cost, the benefit of which is unclear (see point 2).
Answer: measurements of waist circumference were available only in 954 young people of our cohort. We run new comparative analyses in this subsample and demonstrated that waist circumference did not significantly differ between individuals categorized according to the two 1-hour glucose cut-offs. This finding tends to reduce the impact of waist circumference, at least in our sample that was mostly represented by youths with obesity.
4.The authors should clarify in the discussion what is the advantage of performing ONE-HOUR POST-LOAD PLASMA GLUCOSE a in these subjects and compare the cost-effectiveness of this diagnostic protocol with particular reference to social and educational policies for controlling childhood obesity.
Answer: we discussed the above issue in the discussion section. We added the following sentence. “Concerning the cost-effectiveness of performing 1-hour afterload glucose in obese subjects as a screening strategy to identify prediabetes, the data are heterogeneous as the success of a screening program depends on the context (epidemiology, social factors, political priorities and budget constraints). A recent literature review shows that more cost-effective evaluations of national and regional prevention programs for non-communicable diseases are needed to guide policymakers [36].
- The literature suggests that in addition to the blood glucose value after one hour, it is essential to assess the morphology of the curve. Morphologies of the glucose curve seem reflecting different metabolic phenotypes of insulin action and secretion, particularly when combined with morphologies of insulin curve or time of glucose peak. https://eje.bioscientifica.com/view/journals/eje/166/1/107.xml
Answer: we agree with the reviewer that morphologies of glucose curve would have provided more information, but we did not have all the times of the OGTT, apart 60 and 120 min, to assess this further aspect. (Mechanistic Insights into the Heterogeneity of Glucose Response Classes in Youths With Obesity: A Latent Class Trajectory Approach. Tricò et al. Diabetes care 2022) (The Shape of the Glucose Response Curve During an Oral Glucose Tolerance Test: Forerunner of Heightened Glycemic Failure Rates and Accelerated Decline in β-Cell Function in TODAY. Arslanian S, El Ghormli L, Young Kim J, Bacha F, Chan C, Ismail HM, Levitt Katz LE, Levitsky L, Tryggestad JB, White NH; TODAY Study Group. Diabetes Care. 2019 Jan;42(1):164-172
- Beware of plagiarised parts in the paper and whether all 14 authors actually participated in the work
Answer: all the Authors are affiliated to the Italian Society for Pediatric Endocrinology and Diabetology and actively participated to the publication of several papers on cardiometabolic risk in children and adolescents with obesity.

Round 2
Reviewer 3 Report
The authors have made few changes to the paper.
Many critical issues remain.
1. Plagiarism among metods is high
2. The data are similar to others presented in the past by the same group
3. The number of authors is exorbitant, it is not clear what the role of each author was
4. The clinical and therapeutic utility of identifying the performance of G60 cut-off ≥133 mg/dL and ≥155 mg/dL to detect IGT, impaired insulin and β-cell function is unclear. It seems like an extra cost for the patient and that's all.
Author Response
We implemented the introduction and added 7 more references
- Plagiarism among methods is high
Answer: since these articles come from a single database, a high rate of similarity in materials and methods is inevitable. However, we tried to slightly modify the text as much as it was possible. We have cited the papers that represent the source of the described methodology.
- The data are similar to others presented in the past by the same group
Answer: see the comments above
- The number of authors is exorbitant; it is not clear what the role of each author was
Answer: this and the previous studies have been conducted in collaboration with the main Italian centers of pediatric endocrinology which manage obese youths. The large size of the sample, the peculiarity of the OGTT that included also G60, justify the number of authors. In addition, all the authors have actively provided their intellectual contribution to the discussion of the results
- The clinical and therapeutic utility of identifying the performance of G60 cut-off ≥133 mg/dL and ≥155 mg/dL to detect IGT, impaired insulin and β-cell function is unclear. It seems like an extra cost for the patient and that's all.
Answer: since the paper was intended to be submitted to the special issue of IJERPH on “Diabetes: Screening, Prevention, Diagnosis and Therapy”, we sought to deepen the methodologies useful for the screening of T2DM in childhood, in the light of the most recent evidence. We are aware that there are two positions on prediabetes screening. On the one hand, there is the US Preventive Services Task Force position (Ref 26) which demand to clinicians’ judgement the need of screening, whichever is the diagnostic tool (fasting blood glucose, HbA1c and even more so the OGTT). On the other hand, the position of the ADA and EASD is to recommend the screening in subjects at risk, included children with obesity. It has been reported that G60 level may be is a better alternative to the traditional glucose tests to identify earlier individuals at high risk of diabetes. Since G60 has also the advantage of a less time-consuming marker with a diagnostic equivalence for prediabetes risk, a petition has been raised by experts within the EASD (ref n. 7) to consider G60 above 155 mg/dL to define the altered glucose tolerance in adults. We are convinced that this issue should be addressed also in the pediatric field. Therefore, in an attempt to shed light on this relatively recent topic, we compared two 1-hour blood glucose cut-offs, one derived from adults (155 mg/dL) and the other one proposed for children (133 mg/dL), but scarcely analyzed in this population. We think that our paper may be provide further evidence toward the pediatric cutoff, bearing in mind that additional studies are needed to confirm the advantage of this cut-off in pediatric population.
